# The Perth Empathy Scale: Psychometric Properties of the Polish Version and Its Mental Health Correlates

Paweł Larionow [1,*] and David A. Preece [2,3,4]

1 Faculty of Psychology, Kazimierz Wielki University, 85-064 Bydgoszcz, Poland
2 Faculty of Health Sciences, School of Population Health, Curtin University, Perth, WA 6102, Australia; david.preece@curtin.edu.au
3 School of Psychological Science, The University of Western Australia, Perth, WA 6009, Australia
4 Brain and Behaviour Division, Telethon Kids Institute, Perth, WA 6009, Australia
* Correspondence: pavel@ukw.edu.pl

**Abstract:** The Perth Empathy Scale (PES) is a 20-item self-report questionnaire that assesses people's ability to recognize emotions in others (i.e., cognitive empathy) and vicariously experience other's emotions (i.e., affective empathy), across positive and negative emotions. Originally developed in English, the aim of our study was to introduce the first Polish version of the PES and test its psychometric performance. Our sample was 318 people (184 females, 134 males) with ages ranging from 18 to 77. The factor structure was verified with confirmatory factor analysis. Reliability was tested in terms of internal consistency and test–retest reliability. To explore convergent, divergent, and discriminant validity, we examined relationships between the PES and measures of depression, anxiety, and emotional intelligence. It was shown that the scale was characterized by the intended four-factor solution, thus supporting factorial validity. The internal consistency reliability was also good and test–retest reliability was moderate. The convergent, divergent, and discriminant validity were strong. The clinical importance of assessing affective empathy across both positive and negative emotions was supported. Overall, our results therefore suggest that the Polish version of the PES has strong psychometric performance and clinical relevance as a measure of the multidimensional empathy construct.

**Keywords:** affective empathy; anxiety; cognitive empathy; depression; empathy; negative emotions; positive emotions; psychometric properties; psychopathology; questionnaire

## 1. Introduction

Empathy is a relatively stable multidimensional trait comprising two components: cognitive empathy and affective empathy. While cognitive empathy refers to the ability to recognise and understand other people's emotions, affective empathy refers to the ability to share or experience other people's emotions [1]. As there are positive (e.g., joy) and negative emotions (e.g., anger), these empathy concepts can conceptually be applied to both valence domains [2].

While previous studies have typically been more focused on empathy ability for negative emotions, recent research has indicated that empathy abilities across both positive and negative emotions can play different roles in mental health outcomes [2,3] and social behaviours [4]. The distinction between cognitive and affective empathy is also relevant for understanding the development of psychopathology [5–7]. However, to date, only a few studies have assessed the combined characteristics of empathy ability (i.e., cognitive and affective empathy) for both negative and positive emotions separately (e.g., Ziaei et al. [8]), which is likely in part due to the lack of available measures with this functionality in the field. Thus, to facilitate research on the multidimensional empathy construct, the development of comprehensive self-report questionnaires is crucial.

This study aims to examine the psychometric properties of the 20-item Perth Empathy Scale (PES), which is a tool recently introduced by Brett et al. [1] to assess cognitive and affective empathy abilities across both negative and positive emotions. Our study also aims to use the PES to further examine the links between empathy and mental health outcomes in a general community sample of Polish adults.

Originally created in English [1], the PES consists of four intended subscales: Negative Cognitive Empathy (e.g., *Just by seeing or hearing someone, I know if they are feeling sad*), Positive Cognitive Empathy (e.g., *Just by seeing or hearing someone, I know if they are feeling happy*), Negative Affective Empathy (e.g., *When I see or hear someone who is sad, it makes me feel sad too*), and Positive Affective Empathy (e.g., *When I see or hear someone who is happy, it makes me feel happy too*). Several composite scores can also be derived from theoretically meaningful combinations of the subscales: a General Cognitive empathy score and a General Affective empathy score, as well as total scale score as an overall marker of empathy ability. Items are answered on a 5-point Likert scale, with higher scores indicating higher levels of empathy. The 20 PES items were written to assess the core components of cognitive and affective empathy, with items covering both negative and positive emotions. These items were then administered to several samples and subjected to factor analysis and other tests of psychometric performance [1].

To the best of our knowledge, the original study on the PES by Brett et al. [1] is the only study on the PES's psychometric performance. The developers of the PES evidenced in that study that the PES was a psychometrically sound questionnaire, with all subscale and composite scores having good levels of internal consistency reliability. Its factor structure was best represented by three lower order factors, including a general cognitive empathy factor and separate negative and positive affective factors [1]. Thus, the cognitive empathy items for negative and positive emotions were very highly correlated, converging onto a single factor. This suggests that, within the cognitive empathy domain, people's ability to recognize negative emotions in others may be part of the same factor as their ability to recognize positive emotions. Thus, it may be less essential to separate cognitive empathy by valence, as compared to affective empathy. Brett et al. [1] also provided evidence that the empathy construct measured with the PES was statistically separable from alexithymia, indicating good discriminant validity. In our study, we anticipated that the PES would show good discriminant validity against people's current levels of psychological distress. In the original study, convergent validity of the PES was supported by establishing correlational patterns with other older empathy measures. It was also shown that females tended to report higher empathy levels compared to males [1]. Based on these results, we predicted similar patterns in our study.

In our study, we were also interested in evaluating the predictive ability of PES scores for mental health outcomes. Based on other studies on the relationships between other empathy measures and anxiety or depression symptoms [5,7], we predicted that affective empathy (compared to cognitive empathy) would be a significant and clinically relevant predictor of these psychopathology symptoms in our Polish community sample.

Because the PES is currently only available in English, our aim here was to introduce and validate the first Polish version. We tested its factorial validity, reliability (internal consistency and test–retest), convergent validity, divergent validity, and discriminant validity. Based on theory and the past findings with the English PES [1], we predicted that

1.  The PES would be best characterized by a three-factor (subscale) structure, including a general cognitive empathy factor and separate negative and positive affective empathy factors as this was the structure found by Brett et al. [1]. Though, a four-factor structure with valence-specific factors for cognitive empathy would also be theoretically coherent as valence specificity has often been found for other emotional constructs (e.g., Chan et al. [9], Preece et al. [10]);
2.  The PES scores would have high levels of internal consistency reliability, as well as good test–retest reliability;

3. The PES scores would positively and highly correlate with an established measure of emotional intelligence [11,12] because the ability to recognize emotions in others is an established component of many models of emotional intelligence (e.g., Salovey and Mayer [13]). We also expected the PES scores to slightly positively correlate (chiefly for negative affective empathy) with psychopathology symptoms or not correlate with them (positive affective empathy and cognitive empathy), supporting good convergent and divergent validity, respectively [14]. A correlation particularly for negative affective empathy was expected as this would result in higher levels of negative affect and many psychopathologies are characterized by high levels of negative affect and emotion dysregulation [15];

4. The PES would demonstrate good discriminant validity via its empathy construct being separable (statistically) from people's current levels of distress or psychopathology symptoms;

5. Females would report higher empathy levels compared to males [16]. As previous reviews indicated inconsistent links between empathy and age [14], we had no specific hypotheses on the relationships between the PES scores and age. We were also interested in examining the predictive role of the PES scores in mental health outcomes (i.e., anxiety and depression symptoms).

## 2. Materials and Methods

### 2.1. Procedure

Ethics approval for this study was granted by the Kazimierz Wielki University Ethics Committee (No. 1/13.06.2022). The Declaration of Helsinki Ethical Principles were adhered to. Participants did not receive any reimbursement for participation. All participants provided written informed consent for use of their data.

Data collection for our study took place between April 2022 and April 2023. In the first part of the study, which was conducted online (see review on the Internet trials methodology, Paul et al. [17]), participants were recruited via social media (Instagram and Facebook) with posts featuring a link to an online anonymous survey. The survey used the Google Forms platform. In order to prevent fatigue during the survey, not all respondents completed all the measures.

In the second part of the study, which was conducted using the paper-and-pencil method, participants were recruited at the university and informed about the survey during classes. They completed the PES and the Schutte Self-Report Emotional Intelligence Test (SSEIT; its copyrighted Polish version available only in a paper-and-pencil format), and in order to provide the test–retest analysis of the PES, the participants completed the PES a second time after approximately 5 weeks interval between the first test.

As we examined the test–retest reliability of the PES and the study was anonymous, each participant had a unique code which they had to remember in order to participate in the test–retest part of the study. Not all participants who completed the PES for the first time were able to complete the questionnaire twice because of different reasons (e.g., absence from the university, not remembering their code at retest, etc.). Some data of the second measurement period were judged invalid because of inconsistencies in codes. Due to the above-described reasons, we received 34 valid responses (for the first and second measurements) for calculating the test–retest reliability of the questionnaire. Therefore, the attrition rate was about 46%. As the university sample comprised students of social sciences, we did not include the university sample for our factor analytic study conducted in the general community sample.

### 2.2. Participants

In the first and main part of the study, our sample included 318 adults (184 females and 134 males) with ages ranging from 18 to 77 ($M = 28.00$, $SD = 13.61$) from the general population in Poland. In terms of highest education level, 23.27% had a higher education

degree, 62.89% had secondary education, 6.60% vocational education, and 7.23% primary school level education.

In the second part of the study, conducted in a paper-and-pencil format, our sample included 63 adults (47 females, 13 males, 3 individuals did not indicate their gender) recruited at the university (for the first measurement of the test–retest reliability and for assessing the PES's convergent validity with an established emotional intelligence measure). In the second measurement of the PES (i.e., test–retest examination), as above-mentioned, we received valid data from 34 people (see for details Section 2.1).

*2.3. Translation of Questionnaire*

Three independent translators translated the original English version of the PES into Polish. Based on their translations, a common Polish translation was created. Then, we translated it back into English and this back translation was compared with the original English version of the scale. We discussed potential discrepancies and made minor corrections, resulting in the final Polish version of the PES (see Table S1 in Supplementary Materials).

*2.4. Measures*

Our participants filled out a demographic questionnaire and a short battery of self-report questionnaires. Internal consistency reliability coefficients are displayed in Table 1 for all the study measures.

*The Perth Empathy Scale* (PES) is a 20-item self-report questionnaire designed to measure cognitive and affective empathy across positive and negative emotions [1]. The PES consists of four five-item subscales and three composite scores. The subscales are: Negative Cognitive Empathy, Positive Cognitive Empathy, Negative Affective Empathy, and Positive Affective Empathy. The subscales of the Negative and Positive Cognitive Empathy items can also be combined into a General Cognitive empathy composite, and the negative and positive affective items into a General Affective empathy composite, and all items are combined into a total scale score as an overall marker of empathy ability. The statements are scored on a five-point scale ranging from 1 (*almost never*) to 5 (*almost always*), with higher scores indicating higher levels of empathy. The original English version of the PES has demonstrated acceptable to good internal consistency reliability, with a Cronbach's alpha of $\geq 0.70$ for its subscales and composite scores [1].

*The Patient Health Questionnaire-4* (PHQ-4) by Kroenke et al. [18] is a four-item questionnaire for measuring anxiety and depressive symptoms experienced in the previous two weeks. The questionnaire includes two two-item subscales: anxiety (e.g., *Not being able to stop or control worrying*) and depression (e.g., *Little interest or pleasure in doing things*). The total score of these psychopathology symptoms can also be calculated. Higher scores indicate more severe symptoms. The PHQ-4 uses a four-point Likert scale from 0 (*not at all*) to 3 (*nearly every day*). In this study, we used the Polish version of the PHQ-4, which has strong psychometric properties (e.g., a two-factor structure, expected relationships with other constructs, good test–retest reliability, and acceptable internal consistency reliability with a McDonald's omega of $\geq 0.73$ [19].

*The Schutte Self-Report Emotional Intelligence Test* (SSEIT) developed by Schutte et al. [20] was used to measure participants' level of emotional intelligence, understood as the ability to recognize, understand, and manage emotions (one's own or other people's). The SSEIT has 33 statements (e.g., *I like to share my emotions with others*) which are responded to on a five-point Likert scale from 1 (*strongly disagree*) to 5 (*strongly agree*). Higher scores indicate higher levels of emotional intelligence. The Polish version of the SSEIT by Jaworowska and Matczak [21] has satisfactory psychometric properties, including factorial validity, good internal consistency reliability (a Cronbach's alpha of $\geq 0.82$ for different Polish validation samples), and test–retest reliability.

### 2.5. Analytic Strategy

The statistical analyses in this study were conducted using *Statistica* version 13.3 and the *EFAtools* and *lavaan* statistical packages in *R* software version 4.3. We reported descriptive statistics for all study variables for our sample. No data were missing.

#### 2.5.1. Factor Structure

The factor structure of the PES was explored using confirmatory factor analyses (maximum likelihood estimation with robust standard errors and the Satorra-Bentler scaled test statistic). The goodness-of-fit of the models was evaluated using three fit index values: the comparative fit index (CFI), root mean square error of approximation (RMSEA), and standardized root mean square residual (SRMR). In line with commonly used conventions, values below 0.08 for RMSEA and SRMR and values above 0.90 for CFI were considered to indicate acceptable fit levels [22]. Akaike Information Criterion (AIC) values were also used to directly compare the different factor models of the PES; AIC penalizes for model complexity and lower values indicate a better fitting model [23]. Factor loadings of 0.40 or higher were considered as meaningful loadings on a factor [24].

#### 2.5.2. Internal Consistency and Test–Retest Reliability

We calculated McDonald's omega ($\omega$) and Cronbach's alpha ($\alpha$) values as indexes of internal consistency. A value of $\geq 0.70$ was used as the threshold for acceptable reliability [25].

To assess test–retest reliability, we calculated intraclass correlation coefficients using a two-way mixed method with absolute agreement type [26], comparing PES scores at baseline to PES scores at the 5-week follow-up. We also used paired-sample *t*-tests to supplement this comparison between the two time points.

#### 2.5.3. Convergent and Divergent Validity

Pearson correlations were calculated between PES scores and scores from the PHQ-4 and SSEIT scores.

#### 2.5.4. Discriminant Validity

To assess discriminant validity, we conducted a second-order exploratory factor analysis (principal axis factoring with direct oblimin rotation). The variables included were the four subscale scores of the PES and the two subscale scores of the PHQ-4. We expected that empathy (PES) and psychopathology levels (PHQ-4) would extract as separable factors, thus supporting discriminant validity.

#### 2.5.5. Predictive Validity

To explore whether the PES scores predicted significant variance in anxiety and depression symptoms, we conducted three multiple regression analyses. Age and gender (to control for demographic effects) and all of the PES subscales were used as predictors in the regression model. The criterion variables in each regression were either the anxiety, depression, or total scores from the PHQ-4.

#### 2.5.6. Demographic and Emotion Valence Comparisons

The PES scores of females and males were compared by a *t*-test. The Psychometrica calculator [27] was used to calculate the Cohen's *d* effect size. Pearson correlations between the PES scores and age were calculated. We conducted two paired *t*-tests to compare Negative Cognitive Empathy and Positive Cognitive Empathy, as well as Negative Affective Empathy and Positive Affective Empathy scores in order to examine whether emotion valence influenced the extent of people's empathy ability.

## 3. Results

Table 1 presents descriptive statistics for all of the study variables. The PES was reasonably normally distributed across its items and subscales or composite scores; skewness values ranged from −1.15 to 0.64, whereas kurtosis values ranged from −0.94 to 1.02.

**Table 1.** Descriptive statistics, gender comparisons, McDonald's omega (ω), and Cronbach's alpha (α) values for the study variables.

| Scales/SubScales | Total Sample | | | | | Females | | | Males | | | t | Cohen's d |
|---|---|---|---|---|---|---|---|---|---|---|---|---|---|
| | N | ω (95% CI) | α (95% CI) | M | SD | N | M | SD | N | M | SD | | |
| PES Negative Cognitive Empathy | 318 | 0.90 (0.89; 0.92) | 0.90 (0.88; 0.92) | 18.35 | 4.79 | 184 | 19.18 | 4.49 | 134 | 17.2 | 4.97 | 3.70 *** | −0.421 |
| PES Positive Cognitive Empathy | 318 | 0.90 (0.88; 0.91) | 0.89 (0.87; 0.91) | 18.6 | 4.65 | 184 | 19.27 | 4.45 | 134 | 17.69 | 4.79 | 3.01 ** | −0.344 |
| PES Negative Affective Empathy | 318 | 0.81 (0.77; 0.84) | 0.80 (0.76; 0.83) | 12.69 | 4.25 | 184 | 13.47 | 4.47 | 134 | 11.6 | 3.68 | 3.96 *** | −0.450 |
| PES Positive Affective Empathy | 318 | 0.82 (0.79; 0.85) | 0.83 (0.79; 0.85) | 14.57 | 4.56 | 184 | 14.89 | 4.78 | 134 | 14.13 | 4.21 | 1.46 | −0.167 |
| PES Cognitive Empathy | 318 | 0.95 (0.94; 0.96) | 0.95 (0.94; 0.95) | 36.95 | 9.18 | 184 | 38.45 | 8.71 | 134 | 34.9 | 9.44 | 3.47 ** | −0.393 |
| PES Affective Empathy | 318 | 0.87 (0.84; 0.89) | 0.87 (0.84; 0.89) | 27.25 | 7.81 | 184 | 28.36 | 8.21 | 134 | 25.74 | 6.98 | 2.99 ** | −0.340 |
| PES General Empathy | 318 | 0.93 (0.92; 0.94) | 0.93 (0.92; 0.94) | 64.2 | 14.98 | 184 | 66.8 | 14.79 | 134 | 60.63 | 14.56 | 3.70 *** | −0.420 |
| PHQ-4 Anxiety | 257 | 0.73 (0.66; 0.79) | 0.73 (0.66; 0.79) | 3.29 | 1.84 | 129 | 3.66 | 1.84 | 128 | 2.91 | 1.76 | 3.31 ** | −0.417 |
| PHQ-4 Depression | 257 | 0.81 (0.74; 0.85) | 0.81 (0.75; 0.85) | 3.15 | 2 | 129 | 3.28 | 1.97 | 128 | 3.02 | 2.02 | 1.03 | −0.130 |
| PHQ-4 Total score | 257 | 0.85 (0.81; 0.87) | 0.84 (0.80; 0.87) | 6.44 | 3.49 | 129 | 6.94 | 3.44 | 128 | 5.94 | 3.49 | 2.31 * | −0.289 |
| SSEIT Emotional intelligence | 63 | 0.91 (0.86; 0.93) | 0.90 (0.86; 0.93) | 127.46 | 14.88 | 47 | 126.32 | 14.32 | 13 | 132.38 | 14.02 | −1.357 | 0.425 |

*Note*. PES = Perth Empathy Scale; PHQ-4 = Patient Health Questionnaire-4; SSEIT = Schutte Self-Report Emotional Intelligence Test; 95% CI = 95% confidence interval; * $p < 0.05$; ** $p < 0.01$; *** $p < 0.001$. For *t*-tests, degrees of freedom (df) was 316 for the PES comparisons, and df was 255 for the PHQ-4 ones, as well as 58 for the SSEIT ones. Three individuals did not indicate their gender in an SSEIT sample ($N_{total}$ = 63, with $N_{females}$ = 47 and $N_{males}$ = 13).

### 3.1. Factor Structure

Similar to Brett et al. [1], we examined five theoretically informed factor models of the PES of increasing complexities: (1) a one-factor model where all 20 PES items were specified to load on a general empathy factor; (2) a two-factor correlated model comprising Cognitive Empathy and Affective Empathy factors (i.e., with no valence distinctions made); (3) a three-factor correlated model comprising Positive Cognitive Empathy, Negative Cognitive Empathy, and General Affective Empathy factors (i.e., a valence distinction made only for cognitive empathy); (4) another three-factor correlated model comprising the General Cognitive Empathy, Negative Affective Empathy, and Positive Affective Empathy factors (i.e., a valence distinction made only for affective empathy); (5) a four-factor correlated model comprising Positive Cognitive Empathy, Negative Cognitive Empathy, Positive Affective Empathy, and Negative Affective Empathy factors (i.e., valence distinctions made for both cognitive and affective empathy).

Our confirmatory factor analyses showed that the more complex factor models tended to have better fit (Table 2). As expected, the one-factor and two-factor models had poor fit. The three-factor correlated model that split cognitive empathy by valence but not affective empathy had worse fit than the three-factor correlated model that split affective empathy by

valence but not cognitive empathy. This indicated empirically that distinguishing between valence domains was more important for affective empathy than cognitive empathy.

**Table 2.** Goodness-of-fit indices for the PES models (*N* = 318).

| PES Factor Models | $\chi^2/df$ | CFI | TLI | RMSEA (90% Confidence Interval) | SRMR | AIC |
|---|---|---|---|---|---|---|
| One-factor model | 1043.788/170 | 0.728 | 0.696 | 0.140 (0.132; 0.148) | 0.113 | 16,905.07 |
| Two-factor correlated model: Cognitive Empathy and Affective Empathy factors | 603.089/169 | 0.858 | 0.84 | 0.101 (0.093; 0.110) | 0.066 | 16,411.45 |
| Three-factor correlated model: Positive Cognitive Empathy, Negative Cognitive Empathy, and Affective Empathy factors | 597.491/167 | 0.86 | 0.841 | 0.101 (0.093; 0.110) | 0.064 | 16,403.93 |
| Three-factor correlated model: Cognitive Empathy, Negative Affective Empathy, and Positive Affective Empathy factors | 504.716/167 | 0.89 | 0.875 | 0.090 (0.081; 0.099) | 0.058 | 16,288.32 |
| Four-factor correlated model: Positive Cognitive Empathy, Negative Cognitive Empathy, Positive Affective Empathy, and Negative Affective Empathy factors with five error terms * | 314.714/159 | 0.95 | 0.941 | 0.062 (0.052; 0.072) | 0.052 | 16,055.50 |
| Three-factor correlated model: Cognitive Empathy, Negative Affective Empathy, and Positive Affective Empathy factors as well as five error terms (items 4 & 8, 13 & 14, 1 & 3, 17 & 18, 19 & 20) | 348.922/162 | 0.94 | 0.929 | 0.068 (0.058; 0.077) | 0.057 | 16,097.17 |

*Note.* PES = Perth Empathy Scale; $\chi^2$ = chi-square statistic; *df* = degrees of freedom; CFI = comparative fit index; TLI = Tucker–Lewis index; RMSEA = root mean square error of approximation; 90% CI = 90% confidence intervals; SRMR = standardized root mean square residual; AIC = Akaike information criterion. * Heywood case (covariance matrix of latent variables was not positive definite). The modification identified (five error terms: items 4 & 8, 13 & 14, 1 & 3, 17 & 18, 19 & 20 to co-vary) was included in the four-factor correlated model, which removed the presence of the Heywood case.

In the four-factor model analysis, there was a Heywood case [28] where the covariance matrix of latent variables was not positive definite. We analyzed the modification indices and added five correlated residuals (between items 4 & 8, 13 & 14, 1 & 3, 17 & 18, 19 & 20) into the four-factor model. A theoretical rationale for the addition of these correlated error terms was in the fact that: (1) the 13 & 14, 17 & 18, 19 & 20 item pairs refer to the same specific emotion; (2) and items 4 & 8 refer to happy and amused emotions, respectively, and are part of the same subscale, i.e., Positive Affective Empathy; and (3) items 1 & 3 refer to the same component of empathy, i.e., Cognitive Empathy. The adding of these error terms resolved the Heywood case in the four-factor model, and the fit index values were good.

We also tested the three-factor model endorsed in Brett et al. [1] with these five error terms added, and it had also good fit. However, compared to the four-factor model, the fit indices of this three-factor model were worse and its AIC values were higher, suggesting worse fit for this three-factor solution. Taking into account fit indices and AIC values, we therefore chose the four-factor model (corresponding to the four intended subscales) with five error terms as the best factor solution in our dataset (Figure 1). In this four-factor model, all items loaded well on their intended subscale factor (factor loadings from 0.589 to 0.851, all *ps* < 0.001; see Table 3).

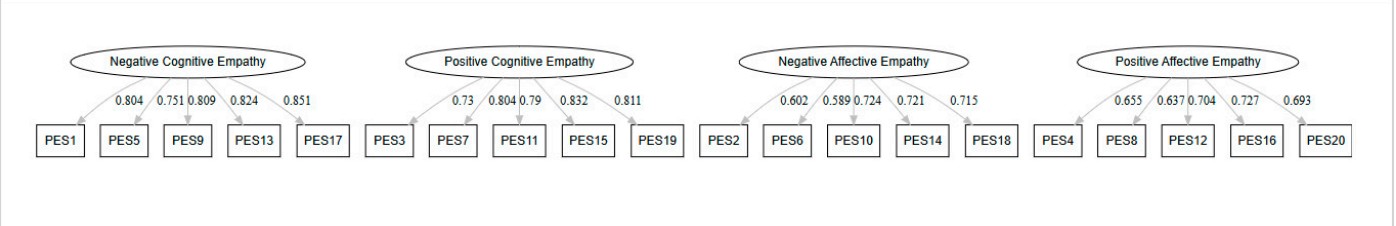

**Figure 1.** Confirmatory factor analysis factor loadings for the 4-factor PES model with five error terms (*N* = 318). All factors in this model were allowed to correlate.

**Table 3.** Completely standardized item factor loadings from confirmatory factor analyses of the 4-factor PES model with five error terms (*N* = 318).

| Subscales | Item Number | Statements | Factor Loadings (All *ps* < 0.001) |
|---|---|---|---|
| Negative Cognitive Empathy | 1 | Just by seeing or hearing someone, I know if they are feeling sad. | 0.804 |
| | 5 | Just by seeing or hearing someone, I know if they are feeling angry. | 0.751 |
| | 9 | Just by seeing or hearing someone, I know if they are feeling scared. | 0.809 |
| | 13 | Just by seeing or hearing someone, I know if they are feeling disgusted. | 0.824 |
| | 17 | Just by seeing or hearing someone, I know if they are feeling embarrassed. | 0.851 |
| Positive Cognitive Empathy | 3 | Just by seeing or hearing someone, I know if they are feeling happy. | 0.73 |
| | 7 | Just by seeing or hearing someone, I know if they are feeling amused. | 0.804 |
| | 11 | Just by seeing or hearing someone, I know if they are feeling calm. | 0.79 |
| | 15 | Just by seeing or hearing someone, I know if they are feeling enthusiastic. | 0.832 |
| | 19 | Just by seeing or hearing someone, I know if they are feeling proud. | 0.811 |
| Negative Affective Empathy | 2 | When I see or hear someone who is sad, it makes me feel sad too. | 0.602 |
| | 6 | When I see or hear someone who is angry, it makes me feel angry too. | 0.589 |
| | 10 | When I see or hear someone who is scared, it makes me feel scared too. | 0.724 |
| | 14 | When I see or hear someone who is disgusted, it makes me feel disgusted too. | 0.721 |
| | 18 | When I see or hear someone who is embarrassed, it makes me feel embarrassed too. | 0.715 |
| Positive Affective Empathy | 4 | When I see or hear someone who is happy, it makes me feel happy too. | 0.655 |
| | 8 | When I see or hear someone who is amused, it makes me feel amused too. | 0.637 |
| | 12 | When I see or hear someone who is calm, it makes me feel calm too. | 0.704 |
| | 16 | When I see or hear someone who is enthusiastic, it makes me feel enthusiastic too. | 0.727 |
| | 20 | When I see or hear someone who is proud, it makes me feel proud too. | 0.693 |

*Note*. PES = Perth Empathy Scale. The English PES items are reproduced here with permission of the copyright holders Brett et al. [1].

The estimated correlation between the Negative Cognitive Empathy and the Positive Cognitive Empathy was positive and very high ($r = 0.97$, $p < 0.001$), and the correlation between the Negative Affective Empathy and the Positive Affective Empathy was 0.73 ($p < 0.001$; Table 4).

**Table 4.** Estimated correlations between the subscales of the 4-factor PES model with five error terms ($N = 318$).

| PES Subscales | Negative Cognitive Empathy | Positive Cognitive Empathy | Negative Affective Empathy |
|---|---|---|---|
| Positive Cognitive Empathy | 0.97 | – | |
| Negative Affective Empathy | 0.5 | 0.49 | – |
| Positive Affective Empathy | 0.5 | 0.65 | 0.73 |

*Note*. PES = Perth Empathy Scale. All estimated correlations are statistically significant (all *ps* < 0.001).

Slightly lower correlations were observed between the Cognitive Empathy subscales and the Affective Empathy subscales, which ranged from 0.49 to 0.65 (all *ps* < 0.001). Our results therefore indicated that, in particular, Negative Cognitive Empathy and Positive Cognitive Empathy are highly correlated, but there is statistical value in separating them.

### 3.2. Internal Consistency and Test–Retest Reliability

The internal consistency reliability of all PES subscales and composite scores was good ($\omega$ and $\alpha \geq 0.80$; see Table 1). Thirty-four participants filled out the PES two times with approximately a 5 week intervals between each test. Intraclass correlation coefficients of all the PES scores between the two-time measurements were moderate (all *ps* < 0.001). For subscale scores, intraclass correlation coefficients were: 0.57 for Negative Cognitive Empathy, 0.50 for Positive Cognitive Empathy, 0.70 for Negative Affective Empathy, and 0.63 for Positive Affective Empathy. For composite scores, intraclass correlation coefficients were: 0.54 for Cognitive Empathy, 0.63 for Affective Empathy, and 0.59 for General Empathy. These results supported moderate test–retest reliability. The paired-samples *t*-test indicated no statistically significant differences on four subscales and three composite scores of the PES between the two time points (all *ps* > 0.05), thus further supporting the test–retest reliability.

### 3.3. Convergent and Divergent Validity

Table 5 presents Pearson correlations between the PES scores and other study variables.

**Table 5.** Pearson correlations between scores on the PES and psychopathology symptoms and emotional intelligence.

| Variables | PHQ-4 Anxiety Symptoms ($N = 257$) | PHQ-4 Depressive Symptoms ($N = 257$) | PHQ-4 Total Score ($N = 257$) | SSEIT Emotional Intelligence ($N = 63$) |
|---|---|---|---|---|
| PES Negative Cognitive Empathy | **0.14 *** | 0.1 | **0.13 *** | **0.53 *** |
| PES Positive Cognitive Empathy | 0.1 | 0.05 | 0.08 | **0.57 *** |
| PES Negative Affective Empathy | **0.25 *** | **0.16 *** | **0.22 *** | −0.08 |
| PES Positive Affective Empathy | −0.02 | −0.09 | −0.06 | **0.58 *** |
| PES Cognitive Empathy | **0.12 *** | 0.08 | 0.11 | **0.60 *** |
| PES Affective Empathy | **0.12 *** | 0.03 | 0.08 | **0.33 *** |
| PES General Empathy | **0.14 *** | 0.06 | 0.11 | **0.61 *** |

*Note*. PES = Perth Empathy Scale; PHQ-4 = Patient Health Questionnaire-4; SSEIT = Schutte Self-Report Emotional Intelligence Test. * $p < 0.05$; ** $p < 0.01$; *** $p < 0.001$. Significant correlations are in bold. The number of the participants (*N*) who completed each questionnaire was shown in the parentheses near the measures.

In general, all of the PES subscale scores (except Negative Affective Empathy) were highly positively correlated with emotional intelligence ($r$ from 0.53 to 0.58, all $ps < 0.001$). All of the PES composite scores were positively correlated with emotional intelligence, with the lowest $r = 0.33$ ($p < 0.01$) for Affective Empathy, to the highest $r = 0.61$ ($p < 0.001$) for General Empathy. Most PES scores (especially Negative Affective Empathy) were moderately positively associated with anxiety symptoms. The PES scores were generally not correlated with depression symptoms, except for a weak positive correlation between Negative Affective Empathy and depression symptoms ($r = 0.16$, $p < 0.05$).

### 3.4. Discriminant Validity

Our second-order exploratory factor analysis of the four PES subscales and the two PHQ-4 subscales (anxiety and depressive symptoms) extracted three factors: Factor 1 we call "cognitive empathy", Factor 2 "psychopathology symptoms", and Factor 3 "affective empathy" (see Table 6).

**Table 6.** Factor loadings from a second-order exploratory factor analysis of the PES subscales and anxiety and depressive symptoms ($N = 257$).

| Variables | Factor 1 ("Cognitive Empathy") | Factor 2 ("Psychopathology Symptoms") | Factor 3 ("Affective Empathy") |
|---|---|---|---|
| PHQ-4 Anxiety symptoms | −0.009 | **0.826** | 0.078 |
| PHQ-4 Depressive symptoms | 0.035 | **0.799** | −0.058 |
| PES Negative Cognitive Empathy | **0.934** | 0.048 | −0.018 |
| PES Positive Cognitive Empathy | **0.958** | −0.028 | 0.031 |
| PES Negative Affective Empathy | −0.013 | 0.158 | **0.727** |
| PES Positive Affective Empathy | 0.089 | −0.180 | **0.767** |
| Proportion of total variance (%) | 43.2 | 22.4 | 7.7 |

*Note*. PES = Perth Empathy Scale; PHQ-4 = Patient Health Questionnaire-4. Factor loadings >0.300 are shown in bold.

As expected, the PES subscales loaded cleanly on the two empathy factors and did not load on the "psychopathology symptoms" factor. These results supported the discriminant validity of the PES.

### 3.5. Predictive Role of Empathy Ability in Anxiety and Depression Levels

Our multiple regression analyses (all models were statistically significant) reinforced that PES scores were significant predictors of psychopathology symptoms (Table 7).

Age and gender explained from 5% to about 6% of the variance in these symptoms. Then, controlling for age and gender, the PES subscale scores were added as predictors, and this led to a significant increase in the explained variance. Beyond age and gender effects, the PES scores explained from 6.3% to 8.4% of the variance in depression symptoms (6.3%), anxiety symptoms (7.5%), or PHQ-4 total scores (8.4%). Negative Affective Empathy was a statistically significant positive predictor of psychopathology symptoms, whereas Positive Affective Empathy was a statistically significant negative predictor of these symptoms. In contrast, cognitive empathy domains were not significant predictors of psychopathology symptoms.

**Table 7.** Regression models for predicting psychopathology symptoms (*N* = 257).

| Predictors | PHQ-4 Anxiety Symptoms | PHQ-4 Depression Symptoms | PHQ-4 Total Score |
|---|---|---|---|
| First step (age and gender as inputted predictors) | | Standardized beta coefficients | |
| Age | −0.16 * | −0.23 *** | −0.21 *** |
| Gender (females = 1, males = 2) | −0.21 *** | −0.08 | −0.16 ** |
| Model parameters | *F*(2, 254) = 8.87, *p* < 0.001 | *F*(2, 254) = 7.72, *p* < 0.001 | *F*(2, 254) = 8.98, *p* < 0.001 |
| Proportion of variance explained (adjusted R$^2$, %) | 5.8 | 5 | 5.9 |
| Second step (age, gender and four PES subscales as inputted predictors) | | Standardized beta coefficients | |
| Age | −0.16 ** | −0.23 *** | −0.21 *** |
| Gender (females = 1, males = 2) | −0.15 * | −0.03 | −0.10 |
| PES Negative Cognitive Empathy | 0.01 | 0.02 | 0.02 |
| PES Positive Cognitive Empathy | 0.08 | 0.1 | 0.1 |
| PES Negative Affective Empathy | 0.34 *** | 0.29 *** | 0.34 *** |
| PES Positive Affective Empathy | −0.26 ** | −0.31 *** | −0.32 *** |
| Model parameters | *F*(6, 250) = 7.54, *p* < 0.001 | *F*(6, 250) = 6.43, *p* < 0.001 | *F*(6, 250) = 8.11, *p* < 0.001 |
| Proportion of variance explained (adjusted R$^2$, %) | 13.3 | 11.3 | 14.3 |
| ΔR$^2$ (%) between the two steps | 7.5 | 6.3 | 8.4 |

Note. PES = Perth Empathy Scale; PHQ-4 = Patient Health Questionnaire-4. * *p* < 0.05; ** *p* < 0.01; *** *p* < 0.001. Significant predictors are in bold.

### 3.6. Demographic and Emotion Valence Comparisons

The Negative Cognitive Empathy, Positive Cognitive Empathy, Negative Affective Empathy, Cognitive Empathy, Affective Empathy, and General Empathy scores of the PES (all effect sizes were small) were higher in females than in males. There were no statistically significant gender differences in Positive Affective Empathy (Table 1).

Pearson correlations between age (skewness = 1.88, kurtosis = 2.47) and the PES scores were calculated. Age was not statistically significantly correlated with any of the PES scores (*r* from −0.02 to 0.10, all *ps* > 0.05).

The participants, on average, reported significantly more Positive Cognitive Empathy (*t*(317) = −2.04, *p* = 0.042, *d* = −0.114, indicating a negligible effect size) and Positive Affective Empathy (*t*(317) = −8.24, *p* < 0.001, *d* = −0.461, indicating a small effect size) levels compared to their Negative Cognitive Empathy and Negative Affective Empathy levels, respectively, indicating some utility of distinguishing emotional valence when assessing both the components of the empathy construct.

### 4. Discussion

Our aim in this study was to introduce the Polish version of the PES and test its validity and reliability. The Polish PES was characterized by a four-factor (subscale) solution, corresponding to the intended subscales. This solution highlights the multidimensional nature of the empathy construct and the importance of considering both the cognitive and affective dimensions, as well as both the negative and positive valence domains. Our factor analytic study highlighted that the Negative Cognitive Empathy and Positive Cognitive Empathy factors were very highly correlated, but there was statistical value in separating them in this sample. We also statistically supported that distinguishing between the valence domains is more important for affective empathy than it is for cognitive empathy, which

is in line with the findings of the PES developers [1]. In Brett et al.'s study [1], separate valence-specific factors were not distinguished for cognitive empathy, which is a slight difference to our study, albeit in our study they were still highly correlated. Thus, it may be that the importance of distinguishing valence in the cognitive empathy domain varies across samples; future work across more diverse samples will be useful to examine the generalizability of these findings. Importantly, our validation study presents the first psychometric evaluation of the PES in a non-English language, thus helping to further understand the structure of empathy cross-culturally.

The internal consistency reliability of the PES, at both the subscale and composite score level, was good ($\omega$ and $\alpha \geq 0.80$), thus supporting that the empathy construct can be robustly measured by the PES at different levels of specificity. These echo the findings of Brett et al. [1] for the English version. The test–retest reliability also appeared to be acceptable, highlighting stability across time for the empathy construct. As our sample for the test–retest analysis was relatively small and this is the first study of the PES to examine stability over time, future studies are needed to examine the test–retest reliability of the PES.

Moreover, our results supported good convergent and divergent validity of the questionnaire. Most of the PES subscales correlated with emotional intelligence scores, suggesting that those with higher empathy also tend to have higher levels of emotional intelligence. Such relationships would be expected given the conceptual overlap between these constructs [11]. Positive and statistically significant correlations between the negatively valenced PES subscales and psychopathology symptoms supported convergent validity (as anxiety and depression are characterised by negative affect), whereas a lack of significant correlations between the positively valenced PES subscales and these symptoms supported a level of divergent validity. The PES showed good discriminant validity against markers of the current level of psychopathology symptoms; this was evident in our factor analysis, supporting that the PES assesses an empathy construct that is statistically separable from one's current levels of psychopathology symptoms or psychological distress.

We noted that our participants tended to report higher empathy abilities for positive emotions as compared to negative emotions, especially in the affective empathy domain. Such findings further reinforce the utility of distinguishing emotional valence when assessing the components of the empathy construct and the need to assess both valence domains. This is consistent with the utility that has been found for valence-specific assessments across a range of other emotional constructs, such as emotion regulation [10,29] and alexithymia [9].

We also examined the predictive role of empathy ability in anxiety and depression symptoms. Controlling for age and gender, our regression models showed that the positive and negative affective empathy domains played important roles as unique predictors of psychopathology symptoms, whereas both cognitive empathy domains did not. Higher levels of negative affective empathy were linked to psychopathology symptoms, whereas positive affective empathy was linked to lower symptoms. As there was a level of specificity in the patterns between two affective empathy components and psychopathology symptoms, our results indicate further support for the clinical relevance of distinguishing emotional valence when assessing the empathy construct.

In general, the results of our study are in line with the previous conclusions on the higher clinical relevance of affective empathy ability and lower relevance of cognitive empathy ability in psychopathology symptoms [5,7]. Importantly, our findings highlight valence-specific relationships. This may be explained by the fact that symptoms like depression and anxiety are characterized by negative affect [30]; as such, if people tend to more easily experience the negative emotions of others, but not the positive emotions of others, this may predispose them to psychopathology. Alternatively, if people more easily experience others' positive emotions, this could be protective in facilitating well-being. Previous studies have shown that different patterns and specificities of cognitive and empathy levels have been observed between clinical samples and health controls [31,32].

As our results are based on the general community sample, it will be important for future work to examine the PES in clinical samples.

We also examined age and gender differences in empathy on the PES. Age was not associated with the PES scores. This suggests that empathy ability, as measured by the PES, is reasonably stable (in adults) across lifespans. That said, this conclusion is tentative because our study is cross-sectional and longitudinal research is needed to examine this pattern empirically. Longitudinal studies on empathy levels across the adult lifespan using other empathy measures have previously supported the idea that empathy is a reasonably stable trait [33]. In terms of gender, most PES subscale and composite scores were higher in females than in males, with small effect sizes for these differences, but no gender differences were noted only in the positive affective empathy scores. Thus, our findings are in line with most previous works in suggesting generally higher empathy abilities in females compared to males [1,34,35].

*Limitations and Future Directions*

Whilst we think this study makes a strong contribution in introducing the first Polish version of the PES and further exploring the structure of the empathy construct, several limitations of the study should be noted. Our sample was recruited from the general community in Poland; however, there was a higher portion of younger people. Our sample for the test–retest analysis was also relatively small. Moreover, this study is cross-sectional; therefore, no casual inferences can be drawn about directionality between empathy and its correlates.

Future studies in more diverse demographic samples, as well as non-clinical and clinical ones, would be beneficial, especially in order to provide group norms for the PES. Future work examining the relationships between the PES and performance-based markers of emotion recognition or empathy would also be beneficial.

## 5. Conclusions

The Polish version of the PES demonstrated good factorial, convergent, divergent, and discriminant validity, as well as good internal consistency and test–retest reliabilities. Overall, the Polish PES appeared to perform similarly to the original English version. The PES therefore seems a strong option for assessments of the empathy construct. Its capacity to assess both the cognitive and affective domains of empathy and do so across the negative and positive valence domains should usefully enable more comprehensive assessments in the field moving forward.

**Supplementary Materials:** The following supporting information can be downloaded at: https://www.mdpi.com/article/10.3390/ejihpe13110182/s1, Table S1: The Polish version of the Perth Empathy Scale.

**Author Contributions:** Conceptualization, P.L.; Data curation, P.L.; Formal analysis, P.L.; Funding acquisition, P.L.; Investigation, P.L.; Methodology, P.L.; Project administration, P.L.; Writing—original draft, P.L.; Writing—review and editing, P.L. and D.A.P. All authors have read and agreed to the published version of the manuscript.

**Funding:** This research received no external funding.

**Institutional Review Board Statement:** The study was conducted in accordance with the Declaration of Helsinki, and the Kazimierz Wielki University Ethics Committee approved the study (No. 1/13 June 2022).

**Informed Consent Statement:** Informed consent was obtained from all subjects involved in the study.

**Data Availability Statement:** The raw data supporting the conclusions of this article are available from the corresponding author on a reasonable request.

**Acknowledgments:** The authors are grateful to Karolina Mudło-Głagolska, Maciej Michalak, and Hanna Pawlicka (Faculty of Psychology, Kazimierz Wielki University) for their help in data collection. The authors are also grateful to the editor and reviewers for their helpful recommendations. The authors would also like to thank the participants of this study for their effort.

**Conflicts of Interest:** The authors declare no conflict of interest.

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
