# Peer review of "The Perth Empathy Scale: Psychometric Properties of the Polish Version and Its Mental Health Correlates"

_ejihpe, doi:10.3390/ejihpe13110182_

Round 1

Reviewer 1 Report

Comments and Suggestions for Authors

Dear Authors,

Thank you for the opportunity to read the article 'The Perth Empathy Scale: Psychometric properties of the Polish Version and its mental health correlates'. The aim of this study was the Polish adaptation of the PES questionnaire and to test its psychometric properties.

The strengths of the manuscript presented for review are the literature cited and the robust validation studies conducted.

The reviewer's job, on the other hand, is to help improve the article so that it meets the highest possible standards of the journal, therefore I will focus on its weaknesses:

Introduction:

[1].  The adapted questionnaire should already be described in the introduction, in particular how it was created, what its dimensions are and how they are defined.

Materials and Methods:

[2].  Sections should be reordered: 2.1 Adaptation procedure, 2.2 Course of study, 2.3 Participants, 2.4 Measures, 2.5 Analytical procedures.

[3].  The rationale is that there are two ways of collecting data - online and paper-and-pencil.

[4].  When describing the questionnaires used in the study, their Cronbach's α reliability coefficients should be given. The Cronbach's α value calculated on the own sample should also be given here.

[5].  In section 2.5, only briefly describe the analytical procedures used in the order in which they were applied and this does not need to be broken down into sections.

[6].  The information contained in lines 180-189 should be transferred to section 3 Results.

Results:

[7].  The relevance and reliability of the adapted tool should be proven first, and only the results (descriptive statistics) should be described.

[8].  The description presented in Section 3.2. is worth enriching with a graphical presentation of the model having the best goodness-of-fit values (this can be done instead of Table 3).

[9].  An explanation is required as to where the tested 2 and 3 factor structure of the scale came from, when the original questionnaire consists of 4 factors.

[10].        An analysis of the hierarchical model would still need to be carried out.

Discussion:

[11].        It would be useful to explicitly present and relate the results of the psychometric analyses obtained by the authors to the results of the original questionnaire, indicating any differences.

General comments:

[12].        No conclusions highlighted in a separate section.

[13].        In a separate section, the limitations of the research should also be highlighted.

[14].        The citations and references should be corrected according to the journal's requirements as described on the page: https://www.mdpi.com/journal/ejihpe/instructions#references

Author Response

We would like to thank the editor and the reviewers for their positive and encouraging feedback on our submission. The constructive comments of the reviewers have helped us to significantly improve the quality of our submission. We have been through all comments one by one, edited the manuscript in detail, and added new material where required. We hope the editor and reviewers find the revised version of the manuscript clear and suitable for publication in the European Journal of Investigation in Health, Psychology and Education. All changes made in the paper are in red.

Dear Authors,

Thank you for the opportunity to read the article 'The Perth Empathy Scale: Psychometric properties of the Polish Version and its mental health correlates'. The aim of this study was the Polish adaptation of the PES questionnaire and to test its psychometric properties.

The strengths of the manuscript presented for review are the literature cited and the robust validation studies conducted.

The reviewer's job, on the other hand, is to help improve the article so that it meets the highest possible standards of the journal, therefore I will focus on its weaknesses:

Introduction:

[1]. The adapted questionnaire should already be described in the introduction, in particular how it was created, what its dimensions are and how they are defined.

Response: We have now added additional detail about the measure’s structure and development into the introduction.

Materials and Methods:

[2]. Sections should be reordered: 2.1 Adaptation procedure, 2.2 Course of study, 2.3 Participants, 2.4 Measures, 2.5 Analytical procedures.

Response: We have now reordered this.

[3]. The rationale is that there are two ways of collecting data - online and paper-and-pencil.

Response: We have now indicated the rationale for this in the paper. Because the Schutte Self-Report Emotional Intelligence Test (SSEIT) is developed and copyrighted by Psychological Tests Laboratory of the Polish Psychological Association, and this test is available only in a pencil-paper format (see https://en.practest.com.pl/sklep/test/INTE), we used two ways of collecting data.

[4]. When describing the questionnaires used in the study, their Cronbach's α reliability coefficients should be given. The Cronbach's α value calculated on the own sample should also be given here.

Response: We presented internal consistency reliability coefficients in our Table 1 as part of the results section. We have now added content within the text of the measures section directing the reader to this table to see these coefficients. We have now also indicated the internal reliability coefficients of the Polish versions of the questionnaires used.

[5]. In section 2.5, only briefly describe the analytical procedures used in the order in which they were applied and this does not need to be broken down into sections.

Response: We would like to keep our descriptions of the analyses applied, because other readers may be interested in these specific statistical considerations. For clarity reasons, we would also like to keep subsections here.

[6]. The information contained in lines 180-189 should be transferred to section 3 Results.

Response: We have now added content in the results section to align with this.

Results:

[7]. The relevance and reliability of the adapted tool should be proven first, and only the results (descriptive statistics) should be described.

Response: We have now reorganized our results section, and we have now presented first the results on factor analysis, reliability, and then presented other analyses.

[8]. The description presented in Section 3.2. is worth enriching with a graphical presentation of the model having the best goodness-of-fit values (this can be done instead of Table 3).

Response: We have now added a figure into the paper showing visually the tested factor model. However, we prefer to also retain Table 3 as we feel the presentation of listing each item with its factor loadings will be of interest to many readers.

[9]. An explanation is required as to where the tested 2 and 3 factor structure of the scale came from, when the original questionnaire consists of 4 factors.

Response: This testing of simpler models allowed us to test the importance of certain conceptual distinctions within the empathy construct. For example, whether it is important to separate between cognitive empathy and affective empathy, or whether it is important to separate between negative and positive emotions within each of these domains. This follows the same methodology employed by Brett et al. (2022) in the development of the PES. We have now added additional content to the introduction and results section to communicate this more clearly.

[10]. An analysis of the hierarchical model would still need to be carried out.

Response: Because our sample size is sufficient for factor analysis, but not large, we felt it was best to focus just on the lower-order solutions, rather than testing for the more complex higher-order models – which typically require large sample sizes for the extra parameters in the model.

Discussion:

[11]. It would be useful to explicitly present and relate the results of the psychometric analyses obtained by the authors to the results of the original questionnaire, indicating any differences.

Response: We have now added content into the discussion section to more comprehensively compare our results to those of the original English PES paper.

General comments:

[12]. No conclusions highlighted in a separate section.

Response: We have now created this section and put key conclusions there.

[13]. In a separate section, the limitations of the research should also be highlighted.

Response: We have now added this extra heading into the discussion section, so as to create a dedicated section on limitations.

[14]. The citations and references should be corrected according to the journal's requirements as described on the page: https://www.mdpi.com/journal/ejihpe/instructions#references

Response: We use free submission format here, acc. to MDPI guidelines, therefore, the reference format can be prepared at the last stage of revisions. Thank you for your efforts in reviewing our paper.

Reviewer 2 Report

Comments and Suggestions for Authors

Thank you very much for the opportunity to review your work. The work is really great. It allows to use the PES scale for many Polish researchers. 

Minor comments: 

-L.114-117 I would suggest to cite some article that used social media in its methodology in preliminary research, it will not raise doubts for other readers, e.g., "I am not a social media expert". 

Korcz, N., Janeczko, E., Bielinis, E., Urban, D., Koba, J., Szabat, P., & Malecki, M. (2021). Influence of informal education in the forest stand redevelopment area on the psychological restoration of working adults. Forests, 12(8), 993. https://doi.org/10.3390/f12080993

Since the study was to test the Polish language version of this test, I think it is possible to include the Polish version of this scale as supplementary material, or an appendix.

The discussion is quite poor in literature. There have already been attempts in Poland and Europe to interpret or translate other psychometric scales into Polish, e.g. Bielinis, E., Jaroszewska, A., Lukowski, A., & Takayama, N. (2020). The effects of a forest therapy program on mental hospital patients with affective and psychotic disorders. International Journal of Environmental Research and Public Health, 17(1), 118. https://doi.org/10.3390/ijerph17145109 

Perhaps this paper will be helpful in interpreting the study more broadly. 

Kindly also change the literature citation throughout the paper according to the journal's editorial requirements - https://www.mdpi.com/journal/ejihpe/instructions

Author Response

We would like to thank the editor and the reviewers for their positive and encouraging feedback on our submission. The constructive comments of the reviewers have helped us to significantly improve the quality of our submission. We have been through all comments one by one, edited the manuscript in detail, and added new material where required. We hope the editor and reviewers find the revised version of the manuscript clear and suitable for publication in the European Journal of Investigation in Health, Psychology and Education. All changes made in the paper are in red.

Comments: Thank you very much for the opportunity to review your work. The work is really great. It allows to use the PES scale for many Polish researchers.

Response: Thank you for these positive comments.

Minor comments:

-L.114-117 I would suggest to cite some article that used social media in its methodology in preliminary research, it will not raise doubts for other readers, e.g., "I am not a social media expert".

Korcz, N., Janeczko, E., Bielinis, E., Urban, D., Koba, J., Szabat, P., & Malecki, M. (2021). Influence of informal education in the forest stand redevelopment area on the psychological restoration of working adults. Forests, 12(8), 993. https://doi.org/10.3390/f12080993

Response: Thank you for your positive feedback on our submission. We recruited our participants via social networks, i.e., Facebook and Instagram, where there was a link to an online anonymous survey by a Google Forms platform with an appended consent form. To provide further context and support for this methodology we have now cited the below-described publication, describing this methodology on collecting data using social media:

Paul, J., Seib, R., & Prescott, T. (2005). The Internet and clinical trials: background, online resources, examples and issues. Journal of medical Internet research, 7(1), e5. https://doi.org/10.2196/jmir.7.1.e5

Comment: Since the study was to test the Polish language version of this test, I think it is possible to include the Polish version of this scale as supplementary material, or an appendix.

Response: Thank you for noticing this. A copy of the scale and its scoring instructions is now provided in the supplementary materials.

Comment: The discussion is quite poor in literature. There have already been attempts in Poland and Europe to interpret or translate other psychometric scales into Polish, e.g. Bielinis, E., Jaroszewska, A., Lukowski, A., & Takayama, N. (2020). The effects of a forest therapy program on mental hospital patients with affective and psychotic disorders. International Journal of Environmental Research and Public Health, 17(1), 118. https://doi.org/10.3390/ijerph17145109

Perhaps this paper will be helpful in interpreting the study more broadly.

Response: We have now expanded our discussion, and at the same time we tried to reach a balance here of providing enough theoretical context, without adding too much information that would change the flow and focus of the paper away from being psychometric.

Comment: Kindly also change the literature citation throughout the paper according to the journal's editorial requirements - https://www.mdpi.com/journal/ejihpe/instructions

Response: We used the free submission format here, acc. to MDPI guidelines, therefore, the reference format can be prepared at the last stage of revisions. Thank you for your efforts in reviewing our paper.

Reviewer 3 Report

Comments and Suggestions for Authors

In my opinion, the article was properly prepared in terms of content and methodology, following the general rules for describing the procedure for adapting a research tool.

Author Response

Thank you for your positive feedback on our submission!

Reviewer 4 Report

Comments and Suggestions for Authors

The authors created a Polish version of the Perth Empathy Scale, which was originally written in English. They provided support for the Polish-translated PES’s reliability, convergent validity with emotional intelligence (EI) and anxiety symptoms, and divergent validity. The authors also used good practices in confirmatory factor analysis to replicate the factor structure found by Brett et al. (2022) in a sample that was relatively diverse in terms of gender, education, and age. Overall, I think the manuscript makes a good contribution to the research.

There are a few major questions and concerns that I had while reading the manuscript, which I will describe below:

1. In the introduction, the authors clearly specify their hypotheses, but the authors could elaborate on the theory underlying these hypotheses. For example, what are some of the specific implications that empathy and its cognitive versus affective and positive versus negative components have on mental health and psychopathology? In what ways is empathy similar and different from EI? What components of empathy are likely to be most relevant to EI? Why do psychopathology symptoms seem especially relevant to the validity of an empathy measure?

2. The authors should provide more detail on the participants and procedures. Specifically, it was initially unclear that the first and second part of the study likely involve different samples, so the authors should specify whether each part of the study consisted of a different sample of participants. That is, how were participants recruited (including compensation) in Facebook/Instagram (e.g., what did the ads consist of), and how were participants recruited in the university? Were there any meaningful differences in age, gender, education, and/or empathy scores between the social media and university participants? How many participants were dropped from the data and why? In the second part of the study, what was the attrition rate from the initial PES to the 5-week follow-up PES?

3. The authors provided evidence for the divergent validity of the PES from anxiety and depression symptoms, but there was no mention of divergence of the PES from EI. The authors could discuss the relationship between the PES and EI in more detail. For example, how consistent is (a) the strength of the correlations between the PES subscales and EI with (b) the overlap between empathy and EI? The authors could also discuss the relatively small number of participants who completed the EI measure.

Author Response

We would like to thank the editor and the reviewers for their positive and encouraging feedback on our submission. The constructive comments of the reviewers have helped us to significantly improve the quality of our submission. We have been through all comments one by one, edited the manuscript in detail, and added new material where required. We hope the editor and reviewers find the revised version of the manuscript clear and suitable for publication in the European Journal of Investigation in Health, Psychology and Education. All changes made in the paper are in red.

Comment: The authors created a Polish version of the Perth Empathy Scale, which was originally written in English. They provided support for the Polish-translated PES’s reliability, convergent validity with emotional intelligence (EI) and anxiety symptoms, and divergent validity. The authors also used good practices in confirmatory factor analysis to replicate the factor structure found by Brett et al. (2022) in a sample that was relatively diverse in terms of gender, education, and age. Overall, I think the manuscript makes a good contribution to the research.

There are a few major questions and concerns that I had while reading the manuscript, which I will describe below:

1. In the introduction, the authors clearly specify their hypotheses, but the authors could elaborate on the theory underlying these hypotheses. For example, what are some of the specific implications that empathy and its cognitive versus affective and positive versus negative components have on mental health and psychopathology? In what ways is empathy similar and different from EI? What components of empathy are likely to be most relevant to EI? Why do psychopathology symptoms seem especially relevant to the validity of an empathy measure?

Response:

Thank you for your positive comments. We have now added additional content to the introduction to make the theoretical rationale behind the hypotheses and relationships between these constructs more clear.

2. The authors should provide more detail on the participants and procedures. Specifically, it was initially unclear that the first and second part of the study likely involve different samples, so the authors should specify whether each part of the study consisted of a different sample of participants. That is, how were participants recruited (including compensation) in Facebook/Instagram (e.g., what did the ads consist of), and how were participants recruited in the university? Were there any meaningful differences in age, gender, education, and/or empathy scores between the social media and university participants? How many participants were dropped from the data and why? In the second part of the study, what was the attrition rate from the initial PES to the 5-week follow-up PES?

Response: Because the Schutte Self-Report Emotional Intelligence Test (SSEIT) is developed and copyrighted by Psychological Tests Laboratory of the Polish Psychological Association, and this test is available only in a pencil-paper format (see https://en.practest.com.pl/sklep/test/INTE), we used two ways of collecting data: online and paper-and-pencil.

There was no reimbursement for participants in the two parts of the study. Our respondents participated voluntarily in this study. This is a common situation in Polish studies when participants decide to participate on their own, without compensation. As for an online part of the study, our link with a survey was posted on researchers' social media pages, and participant were able to see and fill out the survey.

As for participants in the university, they were informed about the survey during the classes. Participants interested in participating in the study were informed about its aim and terms. Next, they provided their consent forms. As we examined test-retest reliability of the PES and the study was anonymous, each participant had a unique code, which they had to remember in order to participate in the study in some weeks. Not all participants from the university who completed the PES for the first time were able to complete the questionnaire twice because of different reasons, i.e., absence from the university, a lack of a code during the second stage of the test-retest study, etc. Some data of the second stage were invalid, because of inconsistencies in codes. In the first stage of the test-retest study, 63 participants were recruited. However, due to above-describe reasons, in the second stage we received 34 valid responses. Therefore, an attrition rate here is about 46%.

Thanks for opportunity to clarify description of our samples. As the sample of the university participants were used only for examining convergent validity (empathy with EI) and test-retest, these participants were not included in our general community sample (n = 318) used for factor analytic study. We have now specified in the paper that we have two samples: main sample (n = 318), and sample for convergent validity/test-retest with n = 63 (in Time 1 measurement where we applied the PES and SSEIT for convergent validity) and with n = 34 after Time 2 (in Time 2 we applied only the PES for examining test-retest). As we conducted our main analyses within our general community sample, we did not examine demographic differences between the general community sample and university sample. As the university sample comprised of students of social sciences, we did not include the university sample for factor analytic study conducted in a general community sample.

3. The authors provided evidence for the divergent validity of the PES from anxiety and depression symptoms, but there was no mention of divergence of the PES from EI. The authors could discuss the relationship between the PES and EI in more detail. For example, how consistent is (a) the strength of the correlations between the PES subscales and EI with (b) the overlap between empathy and EI? The authors could also discuss the relatively small number of participants who completed the EI measure.

Response:

As the SSEIT consisted of statements measuring understanding of others' emotions (e.g., item 18: By looking at their facial expressions, I recognize the emotions people are experiencing; item 25: I am aware of the non-verbal messages other people send; item 26: When another person tells me about an important event in his or her life, I almost feel as though I have experienced this event myself; item 29: I know what other people are feeling just by looking at them; item 32: I can tell how people are feeling by listening to the tone of their voice), we believe that moderate correlations between the PES scores and EI, as measured by SSEIT, seem to be considerable. EI is a broader concept than empathy, as EI supposes understanding one's own emotions and other's emotions as well as managing these emotions.

Thanks for noticing this, as your comment help us to note that only one component of the PES, i.e., the Negative Affective Empathy was not statistically significantly associated with EI (r = -0.08), indicating that experiencing others' negative emotions seems to be useless for people's ability to manage their own and others' emotions. Moreover, the Negative Affective Empathy was a significant predictor of psychopathology symptoms, therefore supporting the theoretical and clinical relevance of the PES not only for clinical psychology, but also for positive psychology with its EI construct.

In our paper-and-pencil study, only 63 participants filled out the EI measure. It is relatively small number of participants, we agree with you. We have now indicated this in the limitations of the study.

Thanks for your helpful and professional comments.

Round 2

Reviewer 1 Report

Comments and Suggestions for Authors

I appreciate the authors' efforts to improve the manuscript. However:

1)     The text contains minor technical errors, e.g. double colons (e.g. line 59), double commas (e.g. line 94). Authors should review the text carefully and correct such shortcomings.

2)     Table 1 with its description should be in section '3.6 Demographic and Emotion Valence Comparisons', not at the beginning of section '3. Results'.

Author Response

Comment: I appreciate the authors' efforts to improve the manuscript. However:

1)     The text contains minor technical errors, e.g. double colons (e.g. line 59), double commas (e.g. line 94). Authors should review the text carefully and correct such shortcomings.

Response: Thank you so much for your positive feedback on our submission. We have now corrected this, and checked the paper for similar issues.

Comment: 2)     Table 1 with its description should be in section '3.6 Demographic and Emotion Valence Comparisons', not at the beginning of section '3. Results'.

Response: Thanks for your comment. We would like to keep this place for Table 1, as this table refers to descriptive statistics, which we present at the beginning of section '3. Results'.  In our experience, tables with descriptive statistics are usually presented at the beginning of results section. Thank you for your time and professional recommendations!